# How to Avoid False-Negative and False-Positive COVID-19 PCR Testing

Irina Fevraleva, Olga Glinshchikova, Tatiana Makarik and Andrey Sudarikov *

National Medical Research Center for Hematology, 125167 Moscow, Russia; fevraleva.i@blood.ru (I.F.); olglin@mail.ru (O.G.); makarik.t@blood.ru (T.M.)

*   Correspondence: dusha@blood.ru; Tel.: +7-(495)613-26-32

**Abstract:** Background: Up to 40% of test results for COVID-19 in the presence of clinical manifestations of the disease might be negative. The reason for a false-negative result might originate from any step of the analysis: poor-quality or empty swab, poor RNA isolation, inactivation of reverse transcriptase or Taq polymerase in the test. Methods: Here we describe a PCR approach for SARS-CoV-2 detection with swab quality and integrity controlled by human *ABL1* mRNA amplification. Designed primers work with the cDNA of the *ABL1* gene, not genomic DNA. Results: The simultaneous appearance of three signals corresponding to the nucleocapsid, spike, and *ABL1* gene indicates infection with the Omicron strain. The amplification of *ABL1* gene and nucleocapsid only indicate other than Omicron infection. The appearance of ABL1 amplification only indicates a true negative result for SARS-CoV-2. All other variants are null and void. Conclusions: A system has been developed for multiplex PCR diagnostics of SARS-CoV-2, which makes it possible to eliminate errors leading to false-negative and false-positive results at all stages of analysis. This is accomplished by the presence of specific primers for human RNA, controlling proper swab application, handling, and all the stages of RT-PCR.

**Keywords:** COVID-19; SARS-CoV-2; RT-PCR

## 1. Introduction

During the current COVID-19 pandemic, the most demanded molecular genetic analysis in the world is a real-time reverse transcription polymerase chain reaction (RT-PCR) for SARS-CoV-2. RT-PCR is the gold standard in the study of any RNA, whether it is the detection of RNA-containing viruses and the determination of viral load or the assessment of the level of gene expression and chimeric transcripts resulting from the fusion of two genes. Despite the high sensitivity of the RT-PCR method, doctors note that about 30–40% of test results for COVID-19 in the presence of clinical manifestations of the disease are negative [1]. A false-negative result of the analysis is the untimely appointment of adequate treatment and dozens of people who came into contact with a non-isolated infected person who received the virus and transmitted it further. A false-negative result may be caused by a poor-quality smear or even an empty swab, poor RNA isolation, or inactivation of reverse transcriptase or Taq polymerase in the sample. Many highly sensitive test systems for the detection of COVID-19 have been developed on the basis of RT-PCR so far. However, positive and negative results obtained when comparing different commercial assays, even those approved by WHO, on the same samples are not always consistent [2,3]. When developing a diagnostic kit, in addition to the search for highly effective primers for detecting SARS-CoV-2, it is very important to choose controls that indicate the successful collection of material, the isolation of virus RNA, and the passage of reverse transcription and PCR reactions. Housekeeping genes mRNA comply as such controls [4,5]. In this study, we propose to use the *ABL1* gene as a housekeeping control system, the RNA of which is detected by the primers we developed in the patient's sample

for SARS-CoV-2. The appearance of the corresponding amplificate of the *ABL1* gene after the completion of RT-PCR indicates the successful isolation of RNA and the passage of all stages of RT-PCR. Only when an amplificate of the *ABL1* gene appears, it is possible to judge the presence or absence of the virus. We used this control system when developing two test systems for COVID-19 multiplex PCR. In one of them, two conserved regions of the SARS-CoV-2 helicase and nucleocapsid genes are simultaneously detected, and in the other, a conserved region of the nucleocapsid gene and the spike gene containing mutations specific for the omicron strain [6,7]. The first test system was developed taking into account the WHO recommendations for diagnostic kits for SARS-CoV-2 RNA to detect two virus genes, and the second allows us to detect SARS-CoV-2 and the current strain ("Omicron" in our case).

## 2. Materials and Methods

Cotton swabs from the nasopharynx or oropharynx of patients admitted to the National Research Center for Hematology were investigated for COVID-19. The swabs were placed in a test tube with 300 µL of phosphate-buffered saline (PBS). About 100 µL of the wash was used to isolate nucleic acids using the "Ribo-prep" kit (Interlabservice, Moscow, Russia) according to the manufacturer's instructions. Reverse transcription combined with PCR was carried out using a set of reagents for reverse transcription (Syntol, Moscow, Russia). The amplification mixture for reverse transcription and PCR (37.5 µL) contained 2.5× buffer, three pairs of primers (10 pmol each), three fluorescently labeled probes (5 pmol each), M-MLV RT reversease (10 units), TAQ polymerase (1 unit) and 20 µL of a solution of nucleic acids isolated from the test sample. Amplification was carried out in a programmable thermostat "CFX96" (BioRad, Hercules, CA, USA) at the following temperature regime: reverse transcription—50 °C for 15 min; —95 °C 5 min., PCR (45 cycles)—95 °C 10 s., 58 °C 10 s., 72 °C 20 s. Sequences of all primers are given in Table 1.

**Table 1.** Primers and fluorescent Taqman probes.

| Gene Target | Primer/ Probe | Sequence 5′-3′ | Ref. |
|---|---|---|---|
| Helicase | Forward Reverse Probe | cgcatacagtcttrcaggct gtgtgatgttgawatgacatggtc FAM-taagatgtggtgcttgcatacgtagac-RTQ2 | [7] |
| Nucleocapsid | Forward Reverse Probe | gcgttcttcggaatgtcg ttggatctttgtcatccaatttg Cy5-aacgtggttgacctacacagst-RTQ2 | [7] |
| Spike (Omicron strain) | Forward Reverse Probe | aacaaaccttgtaatggtgttgc tgctggtgcatgtagaagttc R6G-gatcatatagtttccgacccacttatggtgttggtc-RTQ2 | [6] |
| *ABL1*—internal control | Forward Reverse Probe | gtccacactgcaatgtttttgtg gagttccaacgagcggcttcactc ROX-ccagtagcatctgactttgagcctcag-RTQ2 | This study |

As the negative controls PBS and archival samples stored at −70° since 2010 were used. We considered them as true SARS-CoV-2 negative samples. Threshold PCR cycles obtained with designed control primers were compared to those obtained with primers proposed by the Europe Against Cancer Association (EAC) [8]. For validation same materials were analyzed using One-tube Reverse transcription real-time PCR kit SARS-CoV-2 Cat. No. OOM-136 (Syntol LLC, Moscow, Russia). Sixteen positive samples and 24 negative samples were run in parallel using Syntol kit and proposed set of primers. The results were identical.

## 3. Results

Analysis for SARS-CoV-2 usually consists of four steps. The first step is to collect biological material, the second step is to isolate RNA, the third step is to reverse transcribe isolated RNA, and the fourth step is to amplify cDNA in PCR. When collecting a swab, mucosal cells that contain the patient's RNA, in particular RNA of *ABL1* housekeeping

gene, are necessarily taken into the sample. We use this RNA as an internal control for the completion of all steps including adequate and sufficient collection of the biological material. The design of the primers is such that they can only amplify the cDNA of the *ABL1* gene after reverse transcription of the RNA of the *ABL1* gene, but not its DNA. Primers for the detection of *ABL1* mRNA were chosen considering the exon-intron structure of the gene. Forward and reverse primers corresponded to exonic DNA regions, facilitating RNA detection. The forward primer was chosen from the first exon, reverse from the third exon, and the probe from the seconnd exon of *ABL1* gene. These three consecutive exons in the *ABL1* gene are separated by very long introns and suggested primers flank a region of the *ABL1* gene with a length of more than 139,000 nucleotides thus completely excluding PCR amplification without reverse transcription. However on the transcribed mRNA these exonic sequences are only 104 bases apart and could serve as an effective positive control for the reverse transcription and subsequent amplification.

The use of the *ABL1* gene as a control SARS-CoV-2 RT-PCR detection has been reported previously [9]. However, the primers used in this work flank a region of the *ABL1* gene with a length of less than 700 nucleotides, thus allowing direct PCR amplification without reverse transcription. This setup may not control successful RNA isolation and reverse transcription and therefore can lead to a false-negative SARS-CoV-2 detection.

For greater reliability of the results WHO recommends amplification of two unrelated targets in SARS-CoV-2 RNA. Parallel amplification with two sets of primers should reduce the number of false positive results. However, in a pandemic when laboratories are overloaded, increased time and resource consumption could be critical. Therefore, many commercial SARS-CoV-2 detection systems are targeting one gene only. The solution to this problem is multiplex PCR. Studies on multiplex SARS-CoV-2 PCR detection have shown that neither sensitivity nor specificity of these systems are compromised compared to mono-gene detection approaches, while the probability of obtaining false positive results is greatly reduced [5,6].

SARS-CoV-2 PCR detection was done essentially as described in [7]. However the system was optimized to facilitate multiplex single tube detection with various control primers. Genome positions of SARS-CoV-2 primers are shown on Figure 1.

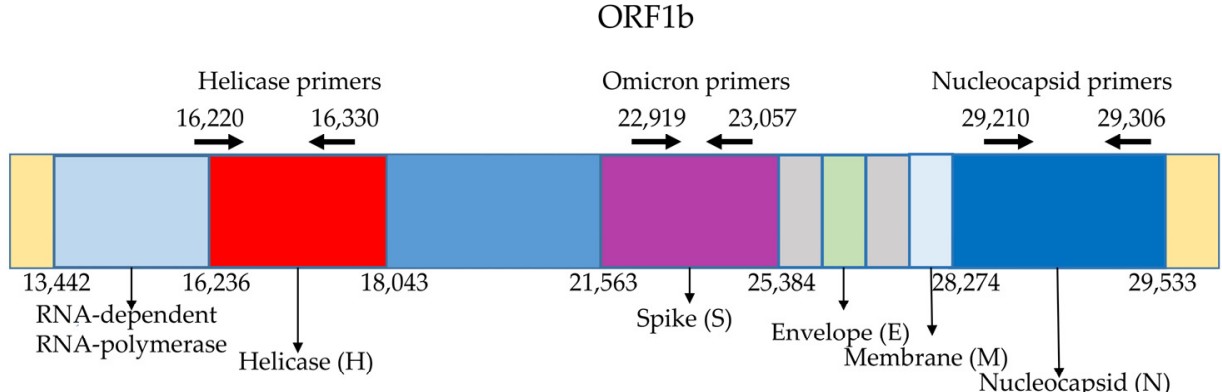

Primers localization in OFR1b of SARS-CoV-2 RNA chain. Numbers denote sequence nucleotide numbers in 5'-3' direction.

**Figure 1.** Genome positions of SARS-CoV-2 primers.

Simultaneous amplification of all three targets corresponding to the cDNA of the helicase gene, the SARS-CoV-2 nucleocapsid, and the human *ABL1* gene indicates infection of the patient with COVID-19. Amplification of *ABL1* gene target alone indicates a true negative result for SARS-CoV-2 (Figure 2a). In 2022, during the Omicron strain outbreak in Russia, we replaced the primers for the helicase gene with primers specific for the Omicron strain in the variable region of the spike gene [4]. In this case, the simultaneous amplification of all three targets indicates that the patient is infected with an Omicron strain, and the appearance of *ABL1* and helicase gene amplifications indicates that the patient is

infected by non-Omicron SARS-CoV-2 strain. The amplification of the *ABL1* gene alone shows that the test is indeed negative for SARS-CoV-2 RNA (Figure 2b). In a multiplex PCR design, depending on the goals, it is worthwhile changing the target in SARS-CoV-2 RNA, but in our opinion, a prerequisite is to use the primer system proposed by us for internal control of the reaction or a similar one.

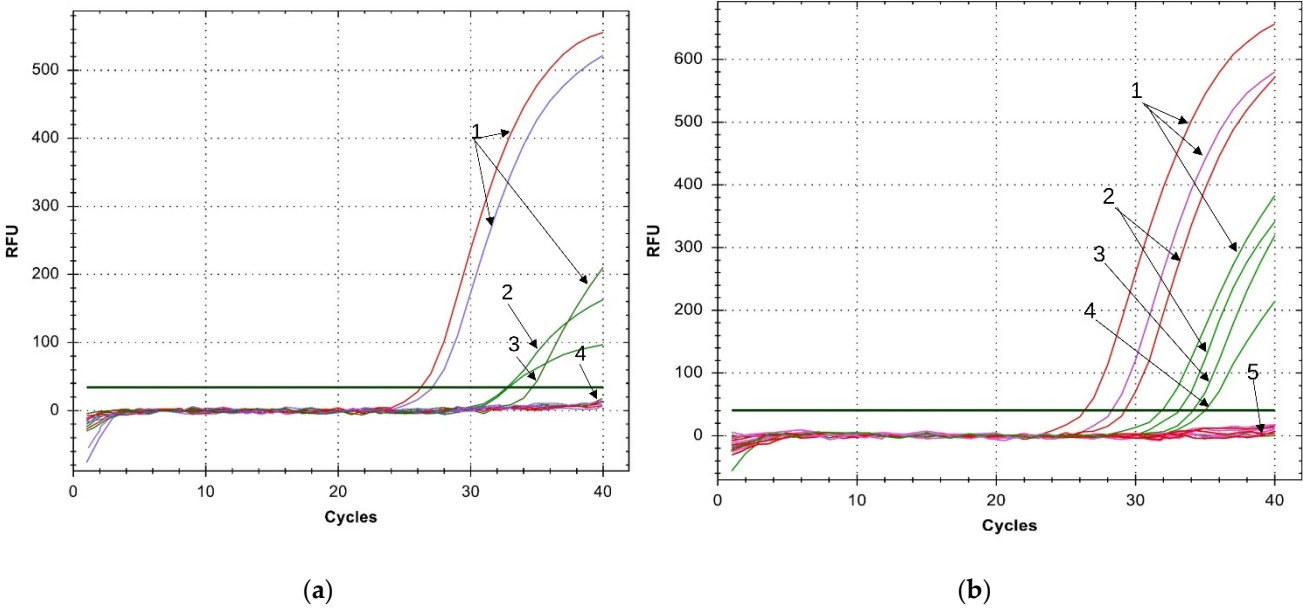

(**a**)  (**b**)

**Figure 2.** Amplification plots: (**a**) red line—helicase, blue line—nucleocapsid, green line—*ABL1*, 1—SARS-CoV-2 positive sample; 2—SARS-CoV-2 negative sample; 3—frozen archival (2010) sample; 4—PBS; (**b**) red line—helicase; purple line—spike (Omicron); green line—*ABL1*, 1—SARS-CoV-2 (Omicron) positive sample, 2—SARS-CoV-2 (non-Omicron) positive sample, 3—SARS-CoV-2 negative sample, 4—frozen archival (2010) sample, 5—PBS.

Threshold cycles of *ABL1* gene amplification with primers recommended by EAC were higher than those obtained with primers suggested by us. This indicates an admixture of genomic DNA in the isolated RNA. Table 2 illustrates the advantage of the proposed *ABL1* gene control over the commercial kit that we used to validate the results. One can see that sample V was of poor quality, and did not contain sufficient RNA material to ensure *ABL1* mRNA amplification. However, a commercial kit gave a negative result, which we prove to be false negative.

**Table 2.** Threshold cycles (Tc) and interpretation of results for commercial (Syntol) kit and proposed set of primers.

| Sample | Helicase Tc | Nucleocapsid Tc | *ABL1* Tc | Syntol Kit Tc | Synlol Kit Control Tc | Our Result | Syntol Kit Result |
|--------|-------------|-----------------|-----------|---------------|------------------------|------------|-------------------|
| N | 21 | 20 | 32 | 23 | 25 | Positive | Positive |
| B | 29 | 28 | 31 | 30 | 24 | Positive | Positive |
| K | >45 | >45 | 32 | >45 | 26 | Negative | Negative |
| V | >45 | >45 | >45 | >45 | 25 | Failure | False-negative |

## 4. Discussion

Since early 2020 the most demanded molecular genetic test is real-time reverse transcription polymerase chain reaction (RT-PCR) for SARS-CoV-2. However, up to 40% of test results for COVID-19 in the presence of clinical manifestations of the disease might be negative. A false-negative analysis means untimely treatment of the patient and the possible spread of infection from a non-isolated person. Despite the fact that there are many highly sensitive RT-PCR test systems for COVID-19 on the market, the search for

new reliable approaches to increase the robustness of testing is still relevant. To control the entire process usually control DNA or an RNA-containing virus (e.g., bacteriophage MS-2) is added to the sample before RNA isolation. This is sufficient to control RNA isolation, reverse transcription, and PCR amplification. However, improper swab application, handling, transporting, or even attempts to falsify the sample, cannot be excluded this way. Here we propose to include primers for the detection of human *ABL1* mRNA to control swab quality and integrity. Designed primers work with the cDNA of the *ABL1* gene, not genomic DNA. Therefore achieved *ABL1* gene amplification ensures that enough material containing host RNA was taken by the swab, and that RNA was nicely isolated, reverse transcribed, and PCR-amplified. We offer primers for the detection of human *ABL1* gene RNA, allowing to control all four steps of the SARS-CoV-2 detection process. These primers are universal, they can also be used for internal control of the PCR detection validity of any RNA viruses, such as SARS-CoV-2, HCV or HIV. We have also successfully used this control primer design in other gene expression studies [10]. The multiplex PCR test system for the detection of SARS-CoV-2, described here is a fast, reliable and efficient method for detecting the virus.

**Author Contributions:** Conceptualization, I.F. and A.S.; investigation, I.F., O.G. and T.M.; writing—original draft preparation, I.F.; writing—review and editing, A.S.; supervision, A.S.; project administration, I.F. All authors have read and agreed to the published version of the manuscript.

**Funding:** This research received no external funding.

**Institutional Review Board Statement:** The study was conducted in accordance with the Declaration of Helsinki, and approved by the Ethics Committee of The National Medical Research Center for Hematology (protocol code 164).

**Informed Consent Statement:** Informed consent was obtained from all subjects involved in the study.

**Data Availability Statement:** The data presented in this study are available on request from the corresponding author.

**Conflicts of Interest:** The authors declare no conflict of interest.

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
