# Peer review of "How to Avoid False-Negative and False-Positive COVID-19 PCR Testing"

_2673-8937, doi:10.3390/ijtm2020018_

Round 1
Reviewer 1 Report
Dear Authors,
The manuscript ”How to avoid false-negative and false-positive covid-19 PCR testing” underlines several analytic and especially pre-analytic steps which can lead to false negative (or false positive) results in diagnostic molecular PCR testing. The approach of using human housekeeping genes as internal extraction and amplification control in a multiplex RT-PCR test setup for SARS-CoV-2 RNA is a simple solution for detecting such errors. Housekeeping genes have been used before, even in commercialized assays (see e.g. Yan et al. 2020, Onwuamah et al. 2021). The use of ABL1 is especially promising for versatility across sample types (oro-, nasopharyngeal swabs, faecal samples, etc.), as the encoded protein, tyrosine kinase, is involved in various cellular processes throughout the human body. The primers and probe could be incorporated into existing or planned in-house multiplex RT-PCR setups. For SARS-COV-2 (variant) detection, in-house PCR setups remain relevant.
The introduction provides adequate context for the study and cites relevant literature, including the sources of primers and probes used for SARS-CoV-2 detection. The methods for the PCR are described in enough detail to be reproducible, but there is no description of if/how you validated the control. You should give sample numbers used for validating the PCR test setup. Did you test a negative control (e.g. PBS without a swab sample for testing the ABL1 control?). The results are described well, but again, a simple (supplementary) table with true and false negatives and positives for detection of the control would be useful. The discussion highlights the usefulness of the ABL1 primers and probe for RT-PCR tests for other viruses.
With kind regards,
Reviewer
Yan, Y., Chang, L., & Wang, L. (2020). Laboratory testing of SARS‐CoV, MERS‐CoV, and SARS‐CoV‐2 (2019‐nCoV): Current status, challenges, and countermeasures. Reviews in medical virology, 30(3), e2106.
Onwuamah, C. K., Okwuraiwe, A. P., Salu, O. B., Shaibu, J. O., Ndodo, N., Amoo, S. O., ... & Audu, R. (2021). Comparative performance of SARS-CoV-2 real-time PCR diagnostic assays on samples from Lagos, Nigeria. Plos one, 16(2), e0246637.
Minor typo / grammar / copy editing suggestions:
Page 1, lines 9-11: Here we describe a PCR approach for SARS-CoV-2 detection with swab quality and integrity controlled by human 10 ABL1 mRNA amplification.
P. 1, l. 35-36: Many highly sensitive test systems for the detection of COVID-19 have been developed on the basis of RT-PCR so far.
P. 3, l. 117-118: Replace “an omicron” with “a SARS-CoV-2 Omicron variant infection”.
P. 3, l. 118-119: The amplification of the ABL1 gene alone shows that the test is indeed negative for SARS-CoV-2 RNA (Figure 119 2b).
Reviewer 2 Report
The article is quite interesting and timely manner. The novelty of the theme is good. The title and abstract of this study matched with the following parts of the article. The methods and materials of the study is valid, reliable, and defined appropriately. The data are presented in an appropriate way. Is it possible to add a diagram to explain the preset findings? The authors are suggested to update references from the most recent and relevant which are appropriate for the study. I believe my observation and subsequent revision might improve the overall quality of this work. Thank you.
